# Semantic Shift Estimation via Dual-Projection and Classifier Reconstruction for Exemplar-Free Class-Incremental Learning

**Run He**[1]  **Di Fang**[1]  **Yicheng Xu**[2]  **Yawen Cui**[3]  **Ming Li**[4]  **Cen Chen**[1]  **Ziqian Zeng**[1]  **Huiping Zhuang**[1]

## Abstract

Exemplar-Free Class-Incremental Learning (EF-CIL) aims to sequentially learn from distinct categories without retaining exemplars but easily suffers from catastrophic forgetting of learned knowledge. While existing EFCIL methods leverage knowledge distillation to alleviate forgetting, they still face two critical challenges: semantic shift and decision bias. Specifically, the embeddings of old tasks shift in the embedding space after learning new tasks, and the classifier becomes biased towards new tasks due to training solely with new data, hindering the balance between old and new knowledge. To address these issues, we propose the Dual-Projection Shift Estimation and Classifier Reconstruction (DPCR) approach for EFCIL. DPCR effectively estimates semantic shift through a dual-projection, which combines a learnable transformation with a row-space projection to capture both task-wise and category-wise shifts. Furthermore, to mitigate decision bias, DPCR employs ridge regression to reformulate a classifier reconstruction process. This reconstruction exploits previous in covariance and prototype of each class after calibration with estimated shift, thereby reducing decision bias. Extensive experiments demonstrate that, on various datasets, DPCR effectively balances old and new tasks, outperforming state-of-the-art EF-CIL methods. Our codes are available at https://github.com/RHe502/ICML25-DPCR.

## 1. Introduction

Class-incremental learning (CIL) (Zhou et al., 2024; Rebuffi et al., 2017) enables models to acquire knowledge from distinct categories arriving sequentially in a task-wise man-

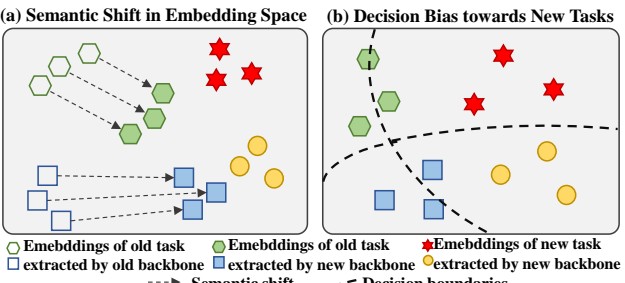

**(a) Semantic Shift in Embedding Space**  **(b) Decision Bias towards New Tasks**

○ Emebddings of old task extracted by old backbone  ⬡ Emebddings of old task extracted by new backbone  ★ Emebddings of new task extracted by new backbone
□ extracted by old backbone  ⬛ extracted by new backbone  ⬤ extracted by new backbone
---▶ Semantic shift  --- Decision boundaries

*Figure 1.* After learning new tasks, (a) the embeddings of old tasks undergo semantic shift in embedding space, (b) and the embeddings are more likely to be classified as new classes.

ner. This paradigm addresses that data is only accessible at specific times or locations, facilitating the accumulation of machine intelligence in dynamic real-world scenarios. However, CIL encounters the challenge of *catastrophic forgetting* (CF) (Belouadah et al., 2021), where the model rapidly loses previous knowledge when acquiring new information. To address forgetting, exemplar-based CIL (EBCIL) (Hou et al., 2019; Zheng et al., 2024; Bian et al., 2024) stores a portion of previous data as exemplars for replaying. However, this method may be limited by privacy concerns or storage capacity constraints. By contrast, Exemplar-free CIL (EFCIL), which does not store previous samples (Rypeść et al., 2024; Masana et al., 2023), is more applicable in practical situations.

Recent EFCIL approaches mainly leverage knowledge distillation to mitigate CF (Li & Hoiem, 2017; Feng et al., 2022; Gao et al., 2024a). However, their performance can be constrained by two key challenges: *semantic shift* in learned representations caused by incremental updates to the backbone, and *decision bias* in classifier training due to the absence of historical data (see Figure 1). These challenges lead to *stability-plasticity dilemma* (Mermillod et al., 2013), where the model struggles to balance preserving old knowledge (stability) with acquiring new knowledge (plasticity). Specifically, when adapting the backbone to new tasks, the embeddings of old classes inevitably shift in the embedding space (Yu et al., 2020), reducing the compatibility of updated representations with previously learned classes and thus compromising stability (Zhu et al., 2021a). Meanwhile, updating the classifier via iterative back-propagation (BP) without access to previous data often results in a prefer-

[1]South China University of Technology [2]Institute of Science Tokyo [3]The Hong Kong Polytechnic University [4]Guangdong Laboratory of Artificial Intelligence and Digital Economy (SZ). Correspondence to: Huiping Zhuang <hpzhuang@scut.edu.cn>.

*Proceedings of the 42nd International Conference on Machine Learning*, Vancouver, Canada. PMLR 267, 2025. Copyright 2025 by the author(s).

ence for the newly learned data, introducing decision bias (Zhuang et al., 2024c; Hou et al., 2019). This bias skews the classification towards new tasks[1], imposing preference on plasticity at the expense of stability.

Existing solutions struggle to simultaneously address both semantic shift and decision bias, resulting in imbalance between stability and plasticity. For instance, several works resort to freezing backbone to prevent alterations of the old representation thus avoid semantic shift (Zhuang et al., 2022b; Goswami et al., 2023; He et al., 2024b). However, this strategy restricts the backbone's ability to adapt representations for new tasks, thereby highly limiting plasticity. To tackle decision bias, several methods adopt a Nearest-Class Mean (NCM) classifier based on prototypes to avoid direct classifier updates (Hou et al., 2019; Goswami et al., 2024; Gomez-Villa et al., 2024). However, as non-parametric classifiers, NCM classifiers heavily depend on the quality of the learned representations, which are susceptible to degradation caused by semantic shift. Also, the absence of trainable parameters could limit their adaptability across task. To address stability-plasticity dilemma, it is imperative to propose a novel EFCIL technique capable of continuously adapting the model to new tasks, effectively addressing semantic shift, and maintaining a well-balanced parametrized classifier.

In this paper, we propose a Dual-Projection Shift Estimation and Classifier Reconstruction (DPCR) approach to handle the semantic shift and decision bias simultaneously. DPCR incorporates a dual-projection (DP) to address the semantic shift, and employs a ridge regression-based classifier reconstruction approach (RRCR) to mitigate the decision bias while benefiting from a parametrized classifier. DP formulates a learnable Task-wise Semantic Shift Projection (TSSP) together with a Category Information Projection (CIP), enabling DPCR to accurately estimate shift and improves stability. RRCR reformulates the classifier training as BP-free classifier reconstruction process using ridge regression. RRCR can leverage previous information encoded in covariance and prototype of each class to obtain a well-balanced classifier with calibration with shift. Our key contributions are summarized as follows:

- We present DPCR, an EFCIL technique that addresses both semantic shift and decision bias, ultimately balancing stability and plasticity in EFCIL.
- To address the semantic shift, we propose the DP to capture the shift. DP comprises a TSSP for learning task-specific shifts and a CIP for offering category-specific information. By comprehensively capturing semantic

shifts across tasks and categories, DP enables DPCR to effectively address semantic shift.
- To address the decision bias, we formulate the RRCR. RRCR establishes a BP-free classifier training framework based on ridge regression, mitigating biases arising from new tasks. When combined with the estimated shift, RRCR yields a less biased classifier.
- Extensive experiments on various benchmark datasets demonstrate that, our DPCR effectively handles semantic shift and decision bias, outperforming the state-of-the-art (SOTA) EFCIL methods with a good balance between stability and plasticity.

## 2. Related Works

EFCIL can be categorized into regularization-based CIL (Li & Zeng, 2023; Zhao et al., 2024; Gao et al., 2024b), prototype-based CIL (Zhu et al., 2021a; Toldo & Ozay, 2022; Wang et al., 2023), and analytic continual learning (Zhuang et al., 2022b; He et al., 2024a; Zeng et al., 2025).

**Regularization-based CIL** imposes additional constraints on activation or key parameters to mitigate forgetting. Knowledge Distillation (KD) is often employed to formulate constraints to mitigate forgetting. For instance, KD can be utilized to restrict activation changes (Li & Hoiem, 2017), establish attention distillation loss (Dhar et al., 2019). Other methods impose constraints on crucial weights from previous tasks (Kirkpatrick et al., 2017). To quantify the importance of parameters, EWC (Kirkpatrick et al., 2017) utilizes a diagonally approximated Fisher information matrix. Various techniques employ different modeling strategies, such as the path integral of loss (Zenke et al., 2017) and changes in gradients (Aljundi et al., 2018). However, the constraints imposed on model updates can hinder the learning of new tasks (Zhuang et al., 2024b).

**Prototype-based CIL** employs prototypes to preserve decision boundaries across old and new tasks. The prototypes are typically the feature means of learned classes and multiple studies have investigated how to reform past feature distributions from prototypes (Petit et al., 2023; Malepathirana et al., 2023; He et al., 2024b). These techniques include prototype augmentations (Zhu et al., 2021b), prototype selection (Zhu et al., 2022) or generating pseudo-features (Rypeść et al., 2024). The reformed distribution is used to guide classifier training to maintain old boundaries. However, due to semantic shift, previous prototypes may become inaccurate. Some techniques address this issue by estimating prototype shift (Shi & Ye, 2023; Malepathirana et al., 2023). For examples, SDC (Yu et al., 2020) utilizes Gaussian kernels to capture the shift by estimating the translation of feature means, ADC (Goswami et al., 2024) uses adversarial samples to improve the accuracy of estimation, and LDC (Gomez-Villa et al., 2024) introduces a learnable

---

[1]This phenomenon of model's tendency to favor new task is also termed as *task-recency bias* (Rebuffi et al., 2017). Several works attribute it to the biased classifier (Zhuang et al., 2024c). We use decision bias to clarify that the bias originates from classifier.

projection to estimate the shift. However, they can obtain less effective estimations since they either only consider the embedding translation (Yu et al., 2020; Goswami et al., 2024) or focus solely on task-wise shift (Gomez-Villa et al., 2024). Moreover, LDC needs iterative BP-training to get the projector, inducing more computation cost. Prototype-based CIL achieves SOTA performance but faces the dilemma of stability and plasticity. That is, the representation is plagued by semantic shift and the classifier may still be biased by replaying previous distribution via BP.

**Analytic continual learning (ACL)** is a recently developed branch of EFCIL that uses closed-form solutions to train the classifier. AL (Guo & Lyu, 2001; Zhuang et al., 2022a; 2021) is a technique that utilizes least squares (LS) to yield closed-form solutions to neural networks training, and ACIL (Zhuang et al., 2022b) first introduces AL to the CIL realm. With a frozen backbone, ACIL is developed by reformulating the CIL procedure into a recursive LS form obtained by solving a ridge regression problem. The ACL family has demonstrated significant performance and has been widely adapted to various scenarios (Zhuang et al., 2023; Fang et al., 2024; Yue et al., 2024; Xu et al., 2024). However, a common issue among existing ACL methods is the limited plasticity due to the frozen backbone. Our DPCR is inspired by ACL and uses the ridge regression to construct a less biased classifier. However, our DPCR does not need to freeze backbone and strikes a good balance between stability and plasticity.

# 3. Proposed Method

In this section, we provide a detailed description of the proposed method. The paradigm of DPCR has three parts: incremental representation learning, shift estimation via dual-projection, and ridge regression-based classifier reconstruction. An overview of DPCR is shown in Figure 2.

## 3.1. Incremental Representation Learning

Prior to further description, we provide definitions of EFCIL here. In EFCIL, the model learns from training data in a task-wise manner. For an EFCIL problem consisting $T$-tasks, in task $t$, the dataset is denoted as $\mathcal{D}_t = \{\mathcal{X}_{t,i}, y_{t,i}\}_{i=1}^{N_t}$, where $\mathcal{X}_{t,i} \in \mathbb{R}^{c \times w \times h}$ is the $i$-th input image in task $t$, $y_{t,i}$ denotes the label of the $i$-th input data, and $N_t$ is the total number of samples in task t. The label set of task $t$ is denoted as $\mathcal{C}_t$ and $\mathcal{C}_i \cap \mathcal{C}_j = \emptyset (i \neq j)$ in EFCIL. In this paper, we focus on the cold-start setting (Magistri et al., 2024), where the model is initialized randomly and the all tasks contains the same number of categories (i.e., $|\mathcal{C}_0| = |\mathcal{C}_j| = C, j \in 1 : T$). In task $t$, the backbone is denoted as $f_{\theta_t} : \mathbb{R}^{c \times w \times h} \to \mathbb{R}^d$ (parameterized by $\theta_t$) and the embeddings extracted by $\theta_t$ is classified via a mapping $\mathbb{R}^d \to \mathbb{R}^{l_t}$, where $d$ is the feature dimension and $l_t = tC$ is the number of classes that the

model have met. The goal of EFCIL is to train the model with $\mathcal{D}_t^{\text{train}}$ at task $t$ without access of data from previous tasks and validate the model on $\mathcal{D}_{1:t}^{\text{test}}$ after training in task $t$.

In task $t$, to adapt the model to new tasks and reduce forgetting, we follow LwF (Li & Hoiem, 2017) to use KD on logits with cross-entropy loss to train the model. In task $t$, the loss function is

$$\mathcal{L}_{\text{rep}} = \mathcal{L}_{\text{ce}}(h_{\tau_t}^{\text{au}}(f_{\theta_t}(\mathcal{X}_t), y_t) + \alpha \mathcal{L}_{\text{kd}}(\mathcal{X}_t), \quad (1)$$

where

$$\mathcal{L}_{\text{kd}} = \mathcal{L}_{\text{ce}}(h_{\tau_{t-1}}^{\text{au}}(f_{\theta_{t-1}}(\mathcal{X}_t)), h_{\tau_t}^{\text{au}}(f_{\theta_t}(\mathcal{X}_t))), \quad (2)$$

$\alpha$ is the weighting parameter of KD, and $h_{t-1}^{au}$, $h_t^{au}$ are auxiliary classifiers in task $t-1$ and $t$ respectively.

Although with KD, training on new task could still leads to semantic shift (Li et al., 2024) and decision bias(Gao et al., 2025). DPCR formulates DP and RRCR to address these issues respectively.

## 3.2. Shift Estimation via Dual-Projection

In this section, we propose the detailed formulation of DP to capture the semantic shift. DP is consisted of a task-wise semantic shift projection (TSSP) to capture the semantic shift between two tasks and a category-wise information projection (CIP) to provide category-related information.

**Task-wise Semantic Shift Projection.** Since we can access the backbone $\theta_{t-1}$ at task $t$, the semantic shift between the backbone $\theta_t$ and $\theta_{t-1}$ can be estimated by a learnable model that captures the difference between embeddings extracted by $\theta_{t-1}$ and $\theta_t$. Inspired by LDC (Gomez-Villa et al., 2024), we introduce a linear projection parameterized by $\boldsymbol{P}^{t-1 \to t} \in \mathbb{R}^{d \times d}$ to transform the embeddings obtained by $\theta_{t-1}$ to those obtained by $\theta_t$.

In task $t$, suppose embeddings of the $i$-th input extracted by the backbone $\theta_t$ and $\theta_{t-1}$ are denoted as $x_{t,i}^{\theta_t} = f_{\theta_t}(\mathcal{X}_{t,i})$ and $x_{t,i}^{\theta_{t-1}} = f_{\theta_{t-1}}(\mathcal{X}_{t,i})$ respectively, and the one-hot label is $\text{onehot}(y_{t,i})$, then the embeddings and one-hot label of all the input data in task-$t$ can be stacked as $\boldsymbol{X}_t^{\theta_{t-1}}$, $\boldsymbol{X}_t^{\theta_t}$, and $\boldsymbol{Y}_t$ respectively:

$$\boldsymbol{X}_t^{\theta_{t-1}} = \begin{bmatrix} \boldsymbol{x}_{t,1}^{\theta_{t-1}} \\ \boldsymbol{x}_{t,2}^{\theta_{t-1}} \\ \vdots \\ \boldsymbol{x}_{t,N_1}^{\theta_{t-1}} \end{bmatrix}, \quad \boldsymbol{X}_t^{\theta_t} = \begin{bmatrix} \boldsymbol{x}_{t,1}^{\theta_t} \\ \boldsymbol{x}_{t,2}^{\theta_t} \\ \vdots \\ \boldsymbol{x}_{t,N_1}^{\theta_t} \end{bmatrix}, \quad \boldsymbol{Y}_t = \begin{bmatrix} \text{onehot}(y_{t,1}) \\ \text{onehot}(y_{t,2}) \\ \vdots \\ \text{onehot}(y_{t,N_1}) \end{bmatrix}. \quad (3)$$

Then the objective is to obtain a projection that satisfies $X_t^{\theta_t} = X_t^{\theta_{t-1}} \boldsymbol{P}^{t-1 \to t}$. We encode the information of semantic shift in $\boldsymbol{P}^{t-1 \to t}$ by minimizing the difference between these two terms. In task t, the optimization problem is

$$\underset{\boldsymbol{P}^{t-1 \to t}}{\arg\min} \mathcal{L}_{\text{mse}} = \|\boldsymbol{X}_t^{\theta_t} - \boldsymbol{X}_t^{\theta_{t-1}} \boldsymbol{P}^{t-1 \to t}\|_{\text{F}}^2, \quad (4)$$

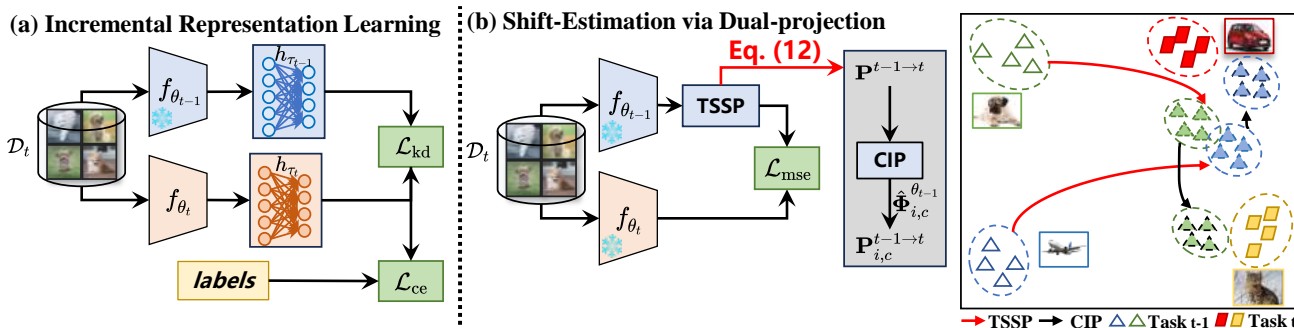

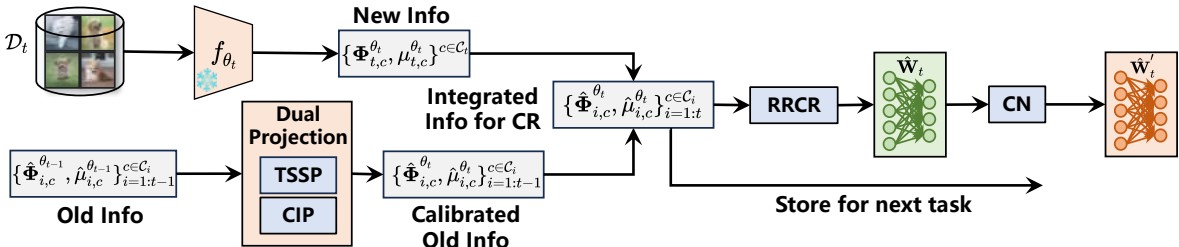

*Figure 2.* An overview of our proposed DPCR. (a) At task t, the backbone is first trained with new data to learn new representation. (b) After the representation learning, shift estimation is conducted with dual-projection (DP), which is consisted of TSSP and CIP. (c) With the DP, the RRCR reconstructs the classifier based on calibrated covariance and prototypes.

where $\|\cdot\|_F$ is Frobenius-norm. The optimal solution to Eq. (4) is $\hat{P}^{t-1\to t} = X_t^{\theta_{t-1}\dagger} X_t^{\theta_t}$, where $\cdot^\top$ denotes transpose operation and $\|\cdot\|^\dagger$ is the Moore-Penrose (MP) inverse (also referred as generalized inverse or pseudo-inverse) (Guo & Lyu, 2001). If $X_t^{\theta_{t-1}}$ is of full-column rank, $X_t^{\theta_{t-1}\dagger} = (X_t^{\theta_{t-1}\top} X_t^{\theta_{t-1}})^{-1} X_t^{\theta_{t-1}\top}$. Here we approximate the MP-inverse by adding a negligible term $\epsilon = 10^{-9}$ to avoid ill-matrix condition, then the projector can be obtained as

$$P^{t-1\to t} = (X_t^{\theta_{t-1}\top} X_t^{\theta_{t-1}} + \epsilon I)^{-1} X_t^{\theta_{t-1}\top} X_t^{\theta_t}, \quad (5)$$

where $I$ is the identity matrix.

By learning the projection, we can model the shift between two tasks and calibrate the previous embeddings. That is, given $X_{t-1}^{\theta_{t-1}}$, we can estimate $\hat{X}_{t-1}^{\theta_t} = X_{t-1}^{\theta_{t-1}} P^{t-1\to t}$. However, TSSP does not consider specificity of each category, and it can be sub-optimal since all the categories within a task are different but share a same estimation of shift. Here we provide a toy case in Figure 2 (b) to illustrate this issue. For task-$t-1$ that contains "dog" and "plane", TSSP can estimate the shift to task t and the embeddings of "dog" and "plane" can be calibrated. The calibrated embeddings can move to the central position in space spanned by the embeddings of task-$t$ due to the objective of task-wise error in Eq. (4). However, this process is sub-optimal since the "dog" is more similar to "cat" ideally. To further separate the calibrated embeddings, we should provide category-related information. We present the CIP to address this issue.

**Category Information Projection.** CIP formulates a sim-

ple but effective training-free row space projection to provide category-related information. Given the class $c \in C_{t-1}$, rows in $X_{t-1,c}^{\theta_{t-1}}$ (stacked matrix of class c's embeddings) span the row space that contains class $c$-related information. To coordinate the TSSP with category information, we further project $P^{t-1\to t}$ onto each class's row space. However, it is memory-intensive to store $X_{t-1,c}^{\theta_{t-1}}$ due to the large quantity of samples. Here we use the uncentered covariance $\Phi_{t-1,c}^{\theta_{t-1}} = X_{t-1,c}^{\theta_{t-1}\top} X_{t-1,c}^{\theta_{t-1}}$ to construct row space projector since it shares the same row space as $X_{t-1,c}^{\theta_{t-1}}$. By applying singular vector decomposition (SVD) to $\Phi_{t-1,c}^{\theta_{t-1}}$, we have

$$U_{t-1,c}, \Sigma_{t-1,c}, U_{t-1,c}^\top = \text{SVD}(\Phi_{t-1,c}^{\theta_{t-1}}), \quad (6)$$

where $U_{t-1,c} = [U_{t-1,c}^r \ U_{t-1,c}^z]$ and $\Sigma_{t-1,c} = [\Sigma_{t-1,c}^r \ \Sigma_{t-1,c}^z]$. $\Sigma_{t-1,c}^r = \text{diag}(\sigma_1, \sigma_2, ..., \sigma_r)$ contains the non-zero singular values $\sigma_i$ ($i = 0 : r$, $r$ is the rank of $\Phi_{t-1,c}^{\theta_{t-1}}$). $\Sigma_{t-1,c}^z$ contains 0 singular values, and $U_{t-1,c}^r = [u_1, u_2, ..., u_r]$ contains the first $r$ singular vectors that span the row space of $\Phi_{t-1,c}^{\theta_{t-1}}$.

Then we can constructed the projector of CIP as $U_{t-1,c}^r U_{t-1,c}^{r\top}$, and the task-wise projector with category information can be constructed as

$$P_{t-1,c}^{t-1\to t} = P^{t-1\to t} U_{t-1,c}^r U_{t-1,c}^{r\top}. \quad (7)$$

With $P_{t-1,c}^{t-1\to t}$, the category-wise shift can also be captured.

The estimation of class within task $t-1$ can be extended to previous tasks. Without loss of generality, for the class

$c \in \mathcal{C}_i, i \in 1 : t - 1$, the embeddings extracted by $\theta_t$ can be estimated task by task. That is, estimating $\hat{X}_{i,c}^{\theta_{i+1}} = X_{i,c}^{\theta_i} P_{i,c}^{i \to i+1}$ in task $i$ then propagate this process in the subsequent tasks. This process can be formulated as follow,

$$\hat{X}_{i,c}^{\theta_t} = \hat{X}_{i,c}^{\theta_{t-1}} P_{i,c}^{t-1 \to t} = X_{i,c}^{\theta_i} \prod_{j=i}^{t-1} P_{i,c}^{j \to j+1}, \quad (8)$$

where

$$P_{i,c}^{j \to j+1} = P^{j \to j+1} \hat{U}_{i,c}^r \hat{U}_{i,c}^{r\top}, \quad (9)$$

$$\hat{U}_{i,c}^r, \hat{\Sigma}_{i,c}^r, \hat{U}_{i,c}^{r\top} = \text{SVD}(\hat{\Phi}_{i,c}^{\theta_j}), \quad \hat{\Phi}_{i,c}^{\theta_j} = \hat{X}_{i,c}^{\theta_j\top} \hat{X}_{i,c}^{\theta_j}. \quad (10)$$

When implementing DP, the covariance can be calibrated in each task and such accumulative operation in Eq. 8 is not necessarily needed. In our implementation, the estimation is conducted across two adjacent tasks (please refer to Algorithm 1).

### 3.3. Ridge Regression-based Classifier Reconstruction

To address the decision bias, we formulate the RRCR to reconstruct the classifier based on encoded information of previous tasks.

Given the joint learning problems from task 1 to $t$ with dataset $\mathcal{D}_{1:t}^{\text{train}} = \mathcal{D}_1^{\text{train}} \cup \mathcal{D}_2^{\text{train}} \cup ... \cup \mathcal{D}_t^{\text{train}}$, the embeddings after backbone $\theta_t$ and the labels can be stacked as $X_{1:t}^{\theta_t}$ and $Y_{1:t}$. The training of the classifier can be formulated as a ridge regression learning problem as follows

$$\underset{W_t}{\arg\min} \quad \|Y_{1:t} - X_{1:t}^{\theta_1} W_t\|_{\text{F}}^2 + \gamma \|W_t\|_{\text{F}}^2, \quad (11)$$

where $\gamma$ is the regularization factor of ridge regression and $W_t$ is the optimal weight for all the tasks that have seen. The optimal solution to Eq. (11) is

$$\hat{W}_t = (\sum_{i=1}^t X_i^{\theta_t\top} X_i^{\theta_t} + \gamma I)^{-1} \sum_{i=1}^t X_i^{\theta_t\top} Y_i. \quad (12)$$

The Eq. (12) can be further decomposed in category-wise form, that is

$$\hat{W}_t = (\sum_{i=1}^t \sum_{c \in \mathcal{C}_i} X_{i,c}^{\theta_t\top} X_{i,c}^{\theta_t} + \gamma I)^{-1} \sum_{i=1}^t \sum_{c \in \mathcal{C}_i} X_{i,c}^{\theta_t\top} Y_{i,c}$$

$$= (\sum_{i=1}^t \sum_{c \in \mathcal{C}_i} \Phi_{i,c}^{\theta_t} + \gamma I)^{-1} \sum_{i=1}^t \sum_{c \in \mathcal{C}_i} H_{i,c}^{\theta_i}, \quad (13)$$

where

$$H_i^{\theta_t} = \sum_{c \in \mathcal{C}_i} X_{i,c}^{\theta_t\top} Y_{i,c} = \sum_{c \in \mathcal{C}_i} N_c \mu_{i,c}^{\theta_t\top} y_{i,c}, \quad (14)$$

$$\Phi_{i,c}^{\theta_t} = X_{i,c}^{\theta_t\top} X_{i,c}^{\theta_t}, \quad \mu_{i,c}^{\theta_t} = \frac{1}{N_c} \sum_{j=1}^{N_c} x_{i,c,j}^{\theta_t}. \quad (15)$$

Here, $\Phi_{i,c}^{\theta_t}$ is the covariance, $\mu_{i,c}^{\theta_t}$ is the prototype of class c and $N_c$ is the number of samples in class c.

As shown in Eq. (13), the classifier can be constructed via covariance and correlation of each task without directly accessing previous data or embeddings. That is, the previous information are encoded in $\Phi_{i,c}^{\theta_t}$ and $H_{i,c}^{\theta_t}$. The new information $\Phi_{t,c}^{\theta_t}$ and $H_{t,c}^{\theta_t}$ can be calculated directly and the classifier can be reconstructed. The formulation of reconstruction leverages the encoded information of previous tasks, then the classifier is not obtained by BP training with only new data, thereby reducing the bias issue.

However, due to the constrain of EFCIL, the embeddings of previous tasks cannot be extracted by $\theta_t$ then the covariance and correlation can not be constructed directly. What we can directly obtained are the information extracted by $\theta_i$, i.e., $\Phi_{i,c}^{\theta_i}, \mu_{i,c}^{\theta_i}$, and $H_{i,c}^{\theta_i}$ and semantic shift affect the encoded information of previous tasks. Here, we incorporate the shift information estimated by dual-projection in Section 3.2 to calibrate the semantic shift.

Without loss of generality, here we consider the calibration in task $t$ and assume this process has also been applied in previous task. Suppose the old information based on $\theta_{t-1}$ (i.e., $\{\hat{\Phi}_{i,c}^{\theta_{t-1}}, \hat{\mu}_{i,c}^{\theta_{t-1}}\}_{i=1:t-1}^{c \in \mathcal{C}_i}$) are reserved, we can calibrate them with the dual-projection. The information after calibration in task $t$ can be estimated via

$$\hat{\Phi}_{i,c}^{\theta_t} = \hat{X}_{i,c}^{\theta_t\top} \hat{X}_{i,c}^{\theta_t} = P_{i,c}^{t-1 \to t\top} \Phi_{i,c}^{\theta_{t-1}} P_{i,c}^{t-1 \to t}, \quad (16)$$

$$\hat{\mu}_{i,c}^{\theta_t} = \frac{1}{N_c} \sum_{j=1}^{N_c} \hat{x}_{i,c,j}^{\theta_{t-1}} = \mu_{i,c}^{\theta_{t-1}} P_{i,c}^{t-1 \to t}, \quad (17)$$

$$\hat{H}_{i,c}^{\theta_t} = \sum_{c \in \mathcal{C}_i}^C N_c \hat{\mu}_{i,c}^{\theta_i\top} y_{i,c}. \quad (18)$$

With the calibrated information of old classes and covariance and prototypes newly computed in task $t$, the integrated information set for RRCR can be formed by $\{\hat{\Phi}_{i,c}^{\theta_t}, \hat{\mu}_{i,c}^{\theta_t}\}_{i=1:t}^{c \in \mathcal{C}_i} = \{\hat{\Phi}_{i,c}^{\theta_t}, \hat{\mu}_{i,c}^{\theta_t}\}_{i=1:t-1}^{c \in \mathcal{C}_i} \cup \hat{\Phi}_{t,c}^{\theta_t}, \hat{\mu}_{t,c}^{\theta_t}\}^{c \in \mathcal{C}_t}$. Then the classifier is reconstructed by

$$\hat{W}_t = (\sum_{i=1}^{t-1} \hat{\Phi}_i^{\theta_t} + \Phi_t^{\theta_t})^{-1}(\sum_{i=1}^{t-1} \hat{H}_i^{\theta_t} + H_t^{\theta_t}). \quad (19)$$

where

$$\hat{\Phi}_i^{\theta_t} = \sum_{c \in \mathcal{C}_i} \hat{\Phi}_{i,c}^{\theta_t}, \quad \hat{H}_i^{\theta_t} = \sum_{c \in \mathcal{C}_i} \hat{H}_{i,c}^{\theta_t}. \quad (20)$$

$$\Phi_t^{\theta_t} = \sum_{c \in \mathcal{C}_t} \Phi_{t,c}^{\theta_t}, \quad H_t^{\theta_t} = \sum_{c \in \mathcal{C}_t} H_{t,c}^{\theta_t}. \quad (21)$$

With the Eq. (19), we reconstruct the classifier with both rectified old information and knowledge from new categories. The old information after calibration is encoded in

$\hat{\Phi}_i^{\theta_t}$ and $\hat{H}_i^{\theta_t}$. This help adjust the old decision boundary to adapt classifier to new tasks. Also, the new information is included in the form of $\Phi_t^{\theta_t}$ and $H_t^{\theta_t}$. Thus the classifier can both ensure plasticity by including new information and maintain the stability by rectifying old boundary.

For the subsequent learning task, we can store the integrated information set of calibrated covariance and prototypes $\{\hat{\Phi}_{i,c}^{\theta_t}, \hat{\mu}_{i,c}^{\theta_t}\}_{i=1:t}^{c \in \mathcal{C}_i}$ to construct $\hat{\Phi}_{i,c}^{\theta_t}$ and $\hat{H}_{i,c}^{\theta_t}$. The memory consumption is $d^2 + d$ for each class.

During the training of DPCR, only the covariance and prototype $\{\hat{\Phi}_c^{\theta_t}, \hat{\mu}_c^{\theta_t}\}$ need to be stored. Semantic shift estimated by the DP is used to calibrate the old information set (i.e., $\{\hat{\Phi}_{i,c}^{\theta_{t-1}}, \hat{\mu}_{i,c}^{\theta_{t-1}}\}_{i=1:t-1}^{c \in \mathcal{C}_i}$ for task $t-1$) then discarded.

**Category-wise Normalization.** In RRCR, calibrating with DP can introduce numerical imbalance to the classifier since $P_{t-1,c}^{t-1 \to t}$ is not unitary. To address this problem, we propose a simple yet effective category-wise normalization (CN). Given the weight of the classifier $\hat{W}_t = [w_1, w_2, ..., w_{tC}]$, where $w_j$ is the weight vector for $j$-th class, the CN is implemented as

$$\hat{W}_t' = [\frac{w_1}{\|w_j\|_1}, \frac{w_2}{\|w_2\|_2}, ..., \frac{w_{tC}}{\|w_{tC}\|_2}]. \qquad (22)$$

After the learning of Eq. (19), the CN is applied on the classifier. The pseudo-code of learning agenda of our proposed DPCR is summarized as Algorithm 1 in Appendix A.

**Complexity Analysis.** Here, we provide the complexity analysis of DP and RRCR. Suppose $F$ is the FLOPs of the model, the time complexity of DP $\mathcal{O}(FN_t + N_t d^2 + tC(d^3 + C^2 d))$ is the sum of those of the feature extraction $\mathcal{O}(FN_t)$, calculation of the task-wise projection matrix $\mathcal{O}(N_t d^2 + d^3)$, and the rectification of each class $\mathcal{O}(d^3 + C^2 d)$. Similarly, the time complexity of RRCR is $\mathcal{O}(d^2 N_c + dtC + td^2 + d^3)$.

## 4. Experiments

### 4.1. Experiment Setting

**Dataset and CIL Protocol.** We validate DPCR and compared methods on three popular benchmark datasets in EFCIL: CIFAR-100 (Krizhevsky et al., 2009), Tiny-ImageNet (Russakovsky et al., 2015), and ImageNet-100 (Douillard et al., 2020). For the CIL evaluation, we follow the cold-start setting adopted in various EFCIL works (Magistri et al., 2024; Goswami et al., 2024). The model is initialized randomly and all the categories are partitioned into $T$-tasks evenly. We report the results of $T = 10$ and 20. To validate the performance on fine-grained and large-scale datasets, we further provide the results on CUB200 (Wah et al., 2011) and ImageNet-1k (Russakovsky et al., 2015).

**Implementation Details.** All the experiments are conducted with the backbone of ResNet-18 (He et al., 2016)

and we run three times for each methods with three different class orders and report the mean results. We set the regularization factor for ridge regression $\gamma = 200, 2000$ and, $2000$ for CIFAR-100, Tiny-ImageNet, and ImageNet-100 respectively. The implementation details of hyperparameters in compared methods can be found in the Appendix B.

**Evaluation Metrics.** We adopt two metrics for evaluation, including *average incremental accuracy* and *final accuracy*. The overall performance is evaluated by the *average incremental accuracy* $\mathcal{A}_{avg} = \frac{1}{T}\sum_{t=1}^{T}\mathcal{A}_t$, where $\mathcal{A}_t$ indicates the average test accuracy after learning task $t$ obtained by testing the model on $\mathcal{D}_{1:t}^{test}$. The other metric *final accuracy* $\mathcal{A}_f$ measures the model's final-task performance after all training agenda. $\mathcal{A}_f$ is an important metric as it reveals the gap between CIL and joint training, a gap that CIL strides to close.

### 4.2. Comparative Study with State-of-the-art Methods

We evaluate our DPCR and various EFCIL methods using the metrics $\mathcal{A}_{avg}$ and $\mathcal{A}_{last}$ in the cold-start setting for $T = 10$ and $T = 20$. The comparison includes LwF (Li & Hoiem, 2017), SDC (Yu et al., 2020), PASS (Zhu et al., 2021b), ACIL (Zhuang et al., 2022b), FeCAM (Goswami et al., 2023), DS-AL (Zhuang et al., 2024b), ADC (Goswami et al., 2024), and LDC (Gomez-Villa et al., 2024). Among these methods, ACIL and DS-AL are ACL techniques that freeze the backbone after the first task, LwF is a regularization-based approach that utilizes KD as a constraint, while PASS, SDC, FeCAM, ADC, and LDC are prototype-based methods. PASS replays augmented prototypes on the classifier, FeCAM improves upon NCM classifier with a Mahalanobis distance-based metric while keeping the backbone frozen, and the other prototype-based methods estimate the semantic shift of prototypes and employ the NCM classifier. The experimental results are tabulated in the Table. 1. We also provide the results of EBCIL methods in Appendix C.

**Main Results.** As shown in Table 1, for the final accuracy $\mathcal{A}_f$, our DPCR demonstrates competitive performance on various settings of $T$ on all the benchmark datasets. For example, the performance of DPCR on CIFAR-100 outperforms the second best method LDC with the gaps of 3.64% and 2.28% under the setting of $T = 10$ and 20 respectively. Similar leading patterns are also observed on Tiny-ImageNet and ImageNet-100. It is worth mentioned that methods that freeze the backbone (e.g., ACIL, FeCAM, and DS-AL) are ineffective and demonstrate poor performance, whereas simple baselines like LwF and SDC can perform better. This observation highlights the limitations of freezing the backbone. Although freezing backbone can eliminate the semantic shift, it sacrifice the adaptation in new tasks then obtain low performance. In contrast to ACIL

*Table 1.* The last-task accuracy and average incremental accuracy in % of compared methods and our proposed DPCR on three benchmark datasets. The data is reported as average after 3 runs with different class orders. Results in **bold** are the best within the compared methods in the same setting.

| Methods | CIFAR-100 | | | | Tiny-ImageNet | | | | ImageNet-100 | | | |
| | T=10 | | T=20 | | T=10 | | T=20 | | T=10 | | T=20 | |
| | $\mathcal{A}_f$ | $\mathcal{A}_{avg}$ | $\mathcal{A}_f$ | $\mathcal{A}_{avg}$ | $\mathcal{A}_f$ | $\mathcal{A}_{avg}$ | $\mathcal{A}_f$ | $\mathcal{A}_{avg}$ | $\mathcal{A}_f$ | $\mathcal{A}_{avg}$ | $\mathcal{A}_f$ | $\mathcal{A}_{avg}$ |
|---|---|---|---|---|---|---|---|---|---|---|---|---|
| LwF (2017) | 42.60 | 58.51 | 36.34 | 51.52 | 26.99 | 42.92 | 18.80 | 33.05 | 42.25 | 61.23 | 30.11 | 50.40 |
| SDC (2020) | 42.25 | 58.43 | 33.10 | 48.68 | 23.86 | 40.66 | 13.45 | 29.70 | 37.68 | 60.33 | 23.64 | 45.52 |
| PASS (2021b) | 44.47 | 55.88 | 28.48 | 42.65 | 23.89 | 36.82 | 12.50 | 25.38 | 36.52 | 52.02 | 19.59 | 31.55 |
| ACIL (2022b) | 35.53 | 50.53 | 27.22 | 39.58 | 26.10 | 41.86 | 21.40 | 33.60 | 44.61 | 59.77 | 33.05 | 48.58 |
| FeCAM (2023) | 34.82 | 49.14 | 25.77 | 41.21 | 29.83 | 42.19 | 22.69 | 34.48 | 41.92 | 58.21 | 28.64 | 43.04 |
| DS-AL (2024b) | 36.83 | 51.47 | 28.90 | 40.37 | 27.01 | 40.10 | 21.86 | 33.55 | 45.55 | 60.56 | 34.10 | 49.38 |
| ADC (2024) | 46.80 | 62.05 | 34.69 | 52.16 | 32.90 | 46.93 | 20.69 | 36.14 | 46.69 | 65.60 | 32.21 | 52.36 |
| LDC (2024) | 46.60 | 61.67 | 36.76 | 53.06 | 33.74 | 47.37 | 24.49 | 38.04 | 49.98 | 67.47 | 34.87 | 54.84 |
| **DPCR (Ours)** | **50.24**$^{\uparrow 3.64}$ | **63.21**$^{\uparrow 1.54}$ | **38.98**$^{\uparrow 2.22}$ | **54.42**$^{\uparrow 1.36}$ | **35.20**$^{\uparrow 1.46}$ | **47.55**$^{\uparrow 0.18}$ | **26.54**$^{\uparrow 2.05}$ | **38.09**$^{\uparrow 0.05}$ | **52.16**$^{\uparrow 2.18}$ | **67.51**$^{\uparrow 0.04}$ | **38.35**$^{\uparrow 3.48}$ | **57.22**$^{\uparrow 2.36}$ |

and DS-AL, DPCR can take advantage of incremental representation learning with semantic shift estimation, enabling it to achieve strong performance. Compared to methods that estimate prototype shift (e.g., SDC, ADC, and LDC), our DPCR exhibits superior performance, showcasing the effective shift estimation of DP. We further analyze DP and RRCR in the following sections. Regarding $\mathcal{A}_f$, DPCR consistently delivers excellent performance, surpassing the compared methods.

**Evolution Curves of Task-wise Accuracy.** To provide a comprehensive view of the results, we display the evolution curves of task-wise accuracy (i.e., $\mathcal{A}_t, t \in 1 : T$) on all datasets. As depicted in Figure 3, starting with similar accuracy in task 1, DPCR attains the highest performance after learning all tasks.

**Validation on Fine-grained Dataset.** To validate our DPCR on fine-grained dataset, we further include the results on CUB200 (Wah et al., 2011) with T=5 in cold-start setting with the same seed of 1993. As shown in Table 2, our DPCR achieves superior results than other methods. On fine-grained dataset like CUB200, our DPCR can take advantage of class-specific information via CIP, thus performs better in the fine-grained scenario.

*Table 2.* Comparative results on CUB200 with $T = 5$.

| CUB200 (T=5) | $\mathcal{A}_f$ (%) | $\mathcal{A}_{avg}$ (%) |
|---|---|---|
| LwF (Li & Hoiem, 2017) | 25.40 | 36.38 |
| ACIL (Zhuang et al., 2022b) | 21.14 | 33.14 |
| DS-AL (Zhuang et al., 2024b) | 21.28 | 32.36 |
| SDC (Yu et al., 2020) | 24.24 | 36.00 |
| ADC (Goswami et al., 2024) | 28.84 | 39.44 |
| LDC (Gomez-Villa et al., 2024) | 28.70 | 39.09 |
| **DPCR (Ours)** | **29.51** | **39.44** |

**Validation on ImageNet-1k.** To validate the performance on large-scale dataset, we compare DPCR and compared methods on ImageNet-1k (Russakovsky et al., 2015) with T=10 and the results are tabulated in Table 3. We use the same backbone of ResNet-18 and same under the same seed of 1993 to conduct the experiments. The hypeparameters are adopted the same as those in ImageNet-100 experiments. As depicted in Table 3, our DPCR can still outperform the compared methods.

*Table 3.* Comparative results on ImageNet-1k with $T = 10$.

| ImageNet-1k (T=10) | $\mathcal{A}_f$ (%) | $\mathcal{A}_{avg}$ (%) |
|---|---|---|
| LwF (Li & Hoiem, 2017) | 22.01 | 42.40 |
| ACIL (Zhuang et al., 2022b) | 32.28 | 46.61 |
| DS-AL (Zhuang et al., 2024b) | 33.67 | 48.84 |
| ADC (Goswami et al., 2024) | 31.34 | 50.95 |
| LDC (Gomez-Villa et al., 2024) | 35.15 | 53.88 |
| **DPCR (Ours)** | **35.49** | **54.22** |

### 4.3. Ablation Study

**Ablation on Components in DPCR.** We perform an ablation study of DP and CN on CIFAR-100 with $T = 10$. The results are presented in Table 4. Here, the baseline is to use the ridge regression to reconstruct the classifier (i.e., RRCR only) after the incremental representation learning. Task-wise semantic shift projection (TSSP), category information projection (CIP), and category-wise normalization (CN) are incorporated into the RRCR sequentially.

*Table 4.* Ablation study of DPCR on CIFAR-100 with $T = 10$.

| Components | $\mathcal{A}_f$ (%) | $\mathcal{A}_{avg}$ (%) |
|---|---|---|
| **RRCR** | 32.17 | 44.89 |
| **RRCR+TSSP** | 40.86 | 55.76 |
| **RRCR+TSSP+CIP** | 45.56 | 62.15 |
| **RRCR+TSSP+CIP+CN** | **51.04** | **64.44** |

As indicated in Table 4, using RRCR only exhibits low

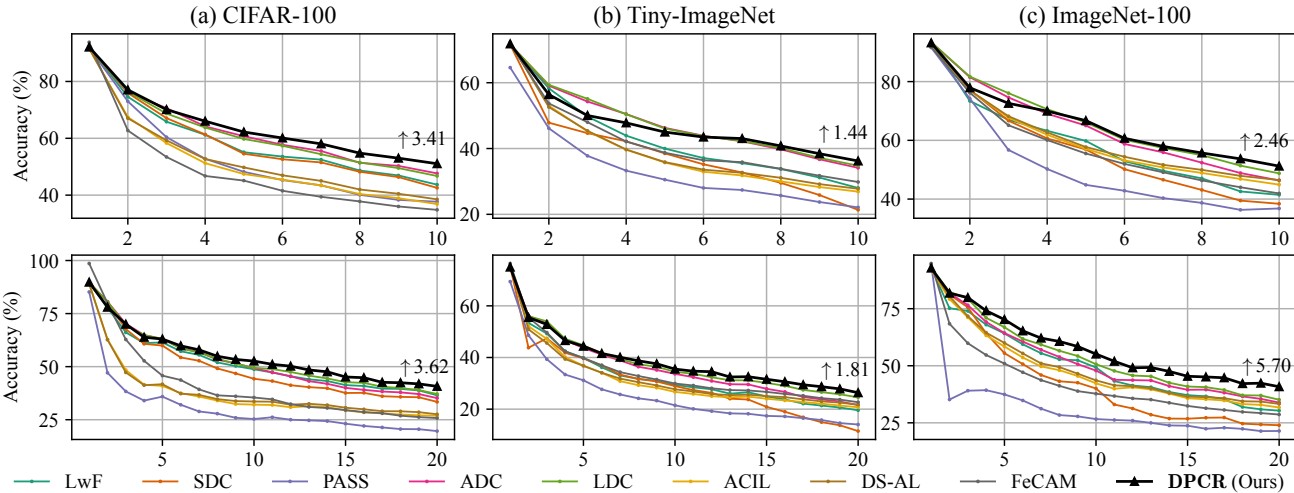

*Figure 3.* Evolution curves of task-wise accuracy.

performance. While RRCR can produce a well-balanced classifier, it could struggle with semantic shift across learning tasks. The inclusion of TSSP leads to a substantial improvement in performance. This highlights that TSSP can mitigate the effects of semantic shift. Furthermore, the addition of CIP results in further performance enhancements, underscoring the efficacy of incorporating projections with category-specific information. By introducing CN, the numerical challenges associated with dual-projection can be mitigated, leading to even better performance for our DPCR.

*Table 5.* Comparison of DP-NCM with other NCM-based methods.

| Methods | CIFAR-100 | | | | Tiny-ImageNet | | | |
| --- | --- | --- | --- | --- | --- | --- | --- | --- |
| | T=10 | | T=20 | | T=10 | | T=20 | |
| | $\mathcal{A}_f$ | $\mathcal{A}_{avg}$ | $\mathcal{A}_f$ | $\mathcal{A}_{avg}$ | $\mathcal{A}_f$ | $\mathcal{A}_{avg}$ | $\mathcal{A}_f$ | $\mathcal{A}_{avg}$ |
| ADC | 47.65 | 62.63 | 35.17 | 52.16 | 30.71 | 41.81 | 18.63 | 31.55 |
| LDC | 47.40 | 62.39 | 37.10 | 53.28 | 32.90 | 43.67 | 23.57 | 34.08 |
| **DP-NCM** | **49.19** | **63.47** | **37.64** | **53.86** | **33.47** | **43.86** | **24.90** | **35.22** |

**DP Effectively Estimates Semantic Shift.** The NCM classifier heavily relies on the backbone's representation and it can be a good performance indicator of semantic shift estimation. As SOTA methods in EFCIL, ADC and LDC estimate the semantic shift of prototypes and utilize an NCM classifier based on calibrated prototypes. To showcase DP's performance in semantic shift estimation, we follow ADC and LDC to leverage an NCM classifier (denoted as DP-NCM) and compare the results. Since the representation learning process remains consistent among the compared methods, we use exactly the same backbone after learning each task and compare the estimation methods fairly. As demonstrated in Table 5, the dual-projection is effective with the NCM classifier, surpassing state-of-the-art methods across various validation scenarios. This superiority can be attributed to the richer information encapsulated in the

estimated shift. That is, both task-wise shift and category-specific details are encoded by linear transformation, while ADC only accounts for embedding translations and LDC solely focuses on task-specific shifts. This observation underscores the effectiveness of dual-projection in semantic shift estimation.

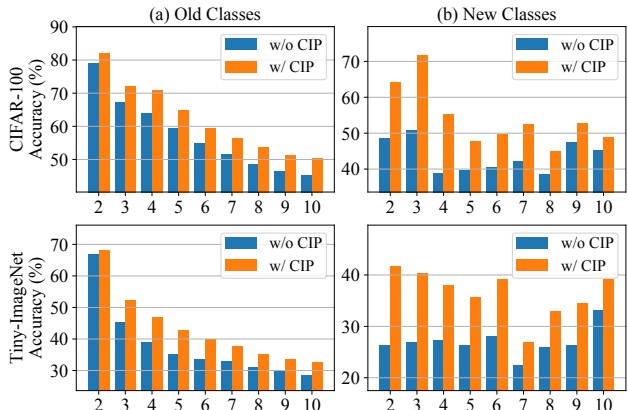

*Figure 4.* Stability-plasticity analysis with and without CIP.

**CIP Enhances Both the Stability and Plasticity.** To illustrate the impact of CIP, we present a visualization of task-wise classification accuracy on CIFAR-100 with $T$=10 for both old (stability) and new classes (plasticity) in Figure 4. Without CIP, the performance on old tasks deteriorates consistently across all tasks and similar pattern of degradation can be also observed on the performance on new classes. This observation highlights the significant influence of CIP on enhancing both stability and plasticity and support the benefit of incorporating category-related information. Furthermore, the results on Tiny-ImageNet align with those on CIFAR-100, demonstrating the robustness of CIP across different datasets.

**Reducing Decision Bias via RRCR.** In this section, we

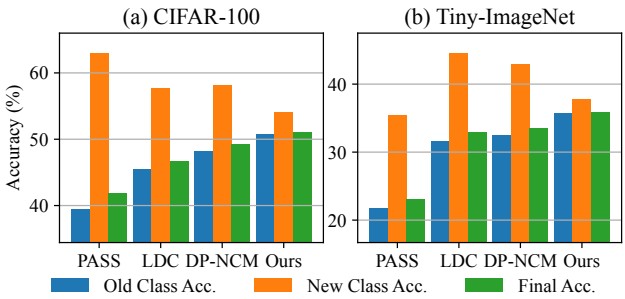

*Figure 5.* Balance Effect of Classification Reconstruction on Stability and Plasticity.

conduct a comparison between different classifier training techniques and showcase the balancing effect of RRCR. We select PASS, LDC, and implement the DP-NCM proposed in section 4.3 as the compared methods. Among these methods, PASS replays augmented prototypes on the classifier, while the other two methods employ the NCM classifier. In Figure 5, we present the accuracy of old classes, new classes, and the final accuracy after a 10-task training on CIFAR-100 and Tiny-ImageNet. These results shows that the accuracy of new classes in the compared methods significantly surpasses that of old classes, whereas our approach demonstrates a less biased pattern of accuracy between old and new classes. Moreover, our performance outperforms DP-NCM, which only differs in classifier training compared with DPCR. These findings indicate that classifier reconstruction can have a superior balancing effect compared to other classifier training techniques, reducing the decision bias. Reconstructing the classifier through ridge regression utilizes accurate information of previous tasks and eliminates the dependency on BP to update the classifier, thereby preventing the direct overwriting of previous decision boundaries with only current data. Additionally, in comparison to NCM, our classifier does not solely rely on the representation thus will not be directly affected by the semantic shift. With a better balance of stability and plasticity, our methods can achieve superior final accuracy.

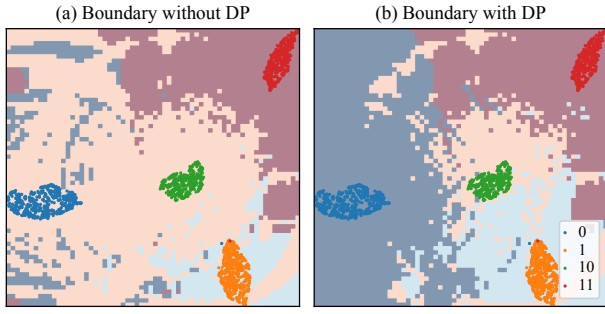

*Figure 6.* DP's impact on decision boundaries viewed after UMAP.

**Visualization of Calibrating Decision Boundaries.** To showcase the impact of the dual-projection in reducing decision bias, we visualize the changes in decision boundaries

with and without DP after the first two tasks on CIFAR-100 using UMAP. As shown in Figure 6, after learning task 2, the decision boundaries are biased to the new classes and become misaligned with the embeddings of class 0. With DP, these decision boundaries can be corrected to accommodate the shifted embeddings. This observation further support DP's effectiveness of semantic shift estimation with RRCR.

## 5. Conclusion

In this paper, we propose the DPCR for EFCIL. To address the semantics shift issue, we proposed the dual-projection (DP) to estimate the semantic shift with a task-wise semantic shift projection (TSSP) and a category information projection (CIP). TSSP estimates the shift across tasks and CIP injects category-related information via row-space projection. To address decision bias, we propose the ridge regression-based classifier reconstruction. The classifier is reconstructed in the form of uncentered covariance and prototypes of each class and the estimated semantic shift can be compensated into the new classifier without accessing previous data. Extensive experiments have been conducted to validate our DPCR and the results demonstrate that, on various datasets, DPCR outperforms state-of-the-art methods with good balancing of stability and plasticity.

## Acknowledgment

This research was supported by the National Natural Science Foundation of China (62306117, 62406114, 62472181), the Guangzhou Basic and Applied Basic Research Foundation (2024A04J3681), GJYC program of Guangzhou (2024D03J0005), National Key R & D Project from Minister of Science and Technology (2024YFA1211500), the Fundamental Research Funds for the Central Universities (2024ZYGXZR074), Guangdong Basic and Applied Basic Research Foundation (2024A1515010220, 2025A1515011413), and South China University of Technology-TCL Technology Innovation Fund.

## Impact Statement

This paper presents work whose goal is to advance the field of Machine Learning. There are many potential societal consequences of our work, none which we feel must be specifically highlighted here.

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

## A. Pseudo-Code of Proposed DPCR

The pseudo-code of the learning agenda of DPCR in task $t$ is proposed in Algorithm 1.

---

**Algorithm 1** Learning Agenda of DPCR in Task $t$

---

**Input:** $\mathcal{D}_t$, $\lambda$, $\theta_{t-1}$, integrated information set of task $t-1$ $\{\hat{\boldsymbol{\Phi}}_{i,c}^{\theta_{t-1}}, \hat{\boldsymbol{\mu}}_{i,c}^{\theta_{t-1}}\}_{i=1:t-1}^{c\in\mathcal{C}_i}$.

**Output:** integrated information set of task $t$ $\{\hat{\boldsymbol{\Phi}}_{i,c}^{\theta_t}, \hat{\boldsymbol{\mu}}_{i,c}^{\theta_t}\}_{i=1:t}^{c\in\mathcal{C}_i}$, new backbone $\theta_t$, and classifier $\hat{\boldsymbol{W}}_t'$.

*// Incremental Representation Learning*

**for** number of epoch **do**

    Update $\theta_t$ by minimizing $\mathcal{L}_{\text{rep}}$ in Eq. (1).

**end for**

*// Shift Estimation via Dual Projection*:

Calculate $\boldsymbol{X}_t^{\theta_{t-1}}$ and $\boldsymbol{X}_t^{\theta_t}$ with $\mathcal{D}_t$, $\theta_{t-1}$ and $\theta_t$.

Obtain task-wise projection $\hat{\boldsymbol{P}}^{t-1\to t}$ by (5).

**for** $i$ in $(i = 1, 2, ..., t-1)$**do**

    **for** $c$ in $\mathcal{C}_i$ **do**

        Obtain $\boldsymbol{U}_{i,c}^r$ by Eq. (6) with $\hat{\boldsymbol{\Phi}}_{i,c}^{\theta_{t-1}}$.

        Obtain $\boldsymbol{P}_{i,c}^{t-1\to t}$ by Eq. (7) with $\boldsymbol{U}_{i,c}^r$.

        *// Rectification*:

        Obtain $\hat{\boldsymbol{\Phi}}_{i,c}^{\theta_t}$ and $\hat{\boldsymbol{\mu}}_{i,c}^{\theta_t}$ by Eq. (16) and Eq. (17)

        Obtain $\hat{\boldsymbol{H}}_{i,c}^{\theta_t}$ via Eq. (18).

    **end for**

    Obtain $\hat{\boldsymbol{\Phi}}_i^{\theta_t} \leftarrow \sum_{c\in\mathcal{C}_i}^{|\mathcal{C}_i|} \hat{\boldsymbol{\Phi}}_{i,c}^{\theta_t}$, $\hat{\boldsymbol{H}}_i^{\theta_t} \leftarrow \sum_{c\in\mathcal{C}_i}^{|\mathcal{C}_i|} \hat{\boldsymbol{H}}_{i,c}^{\theta_t}$.

**end for**

*// Ridge Regression-based Classifier Reconstruction*

**for** $c$ in $\mathcal{C}_t$ **do**

    Obtain $\boldsymbol{\Phi}_{t,c}^{\theta_t} \leftarrow \boldsymbol{X}_{t,c}^{\theta_t\top} \boldsymbol{X}_{t,c}^{\theta_t}$, $\boldsymbol{\mu}_{t,c}^{\theta_t} \leftarrow \frac{1}{N_c} \sum_{j=1}^{N_c} \boldsymbol{x}_{t,c,j}^{\theta_t}$.

    Obtain $\boldsymbol{H}_{t,c}^{\theta_t} = N_c \boldsymbol{\mu}_{t,c}^{\theta_i\top} \boldsymbol{y}_{t,c}$.

**end for**

Calculate $\boldsymbol{\Phi}_t^{\theta_t} \leftarrow \sum_{c\in\mathcal{C}_t}^{|\mathcal{C}_t|} \hat{\boldsymbol{\Phi}}_{t,c}^{\theta_t}$, $\boldsymbol{H}_t^{\theta_t} \leftarrow \sum_{c\in\mathcal{C}_t}^{|\mathcal{C}_t|} \boldsymbol{H}_{t,c}^{\theta_t}$.

Update $\{\hat{\boldsymbol{\Phi}}_{i,c}^{\theta_t}, \hat{\boldsymbol{\mu}}_{i,c}^{\theta_t}\}_{i=1:t}^{c\in\mathcal{C}_i} \leftarrow \{\hat{\boldsymbol{\Phi}}_{i,c}^{\theta_t}, \hat{\boldsymbol{\mu}}_{i,c}^{\theta_t}\}_{i=1:t-1}^{c\in\mathcal{C}_i} \cup \{\hat{\boldsymbol{\Phi}}_{t,c}^{\theta_t}, \hat{\boldsymbol{\mu}}_{t,c}^{\theta_t}\}^{c\in\mathcal{C}_t}$.

Obtain the classifier weight $\hat{\boldsymbol{W}}_t$ by Eq. (19).

Obtain the classifier weight $\hat{\boldsymbol{W}}_t'$ with CN by Eq. (22).

---

## B. Implementation Detail

We mainly follow the implementation details of hyperparameters in ADC (Goswami et al., 2024) for the backbone training. For all the compared methods, we run the experiments three times with non-shuffled class order and two shuffled class orders generated by random seeds of 1992 and 1993. The detailed experimental settings are listed below.

**Data Augmentation.** As implemented in PyCIL (Zhou et al., 2023), for CIFAR-100, we use the augmentation policy which consists of random transformations including contrast or brightness changes. For the other datasets, we use the default setting of augmentations which include random crop and random horizontal flip. We use the same set augmentations for all the methods.

**LwF** (Li & Hoiem, 2017). For LwF, in the first task, we use a starting learning rate of 0.1, momentum of 0.9, batch size of 128, weight decay of 5e-4 and trained for 200 epochs, with the learning rate reduced by a factor of 10 after 60,120, and 160 epochs, on all benchmark datasets. For subsequent tasks, we use an initial learning rate of 0.05 for CIFAR-100 and ImageNet-100 and 0.001 for Tiny-ImageNet. The learning rate is reduced by a factor of 10 after 45 and 90 epochs and the model is trained for a total of 100 epochs. We set the the temperature to 2 and the regularization strength to 10 for CIFAR-100 and Tiny-ImageNet and 5 for ImageNet-100. For the backbone training in SDC, ADC, LDC, ACIL, DS-AL, and our DPCR, we use the same setting as LwF.

**SDC** (Yu et al., 2020). For SDC, we use the $\sigma = 0.3, 0.3, 1.0$ for the Gaussian kernel on CIFAR-100, Tiny-ImageNet, and ImageNet-100 respectively.

**PASS** (Zhu et al., 2021b). For PASS, we tune $\lambda_{\text{proto}}$ from $\{0.1, 0.5, 1.0, 5.0, 10\}$ use $\lambda_{\text{proto}} = 0.1, 0.1, 10$ for CIFAR-100, Tiny-ImageNet, and ImageNet-100 respectively. We follow the implementation in PyCIL (Zhou et al., 2023) to use $\lambda_{fkd} = 10$.

**ADC** (Goswami et al., 2024). For ADC, we follow the implementation in ADC paper to use $\alpha = 25, i = 3$, and $m = 100$ on all the benchmark datasets.

**LDC** (Gomez-Villa et al., 2024). For LDC, we follow the official implementation to use Adam optimizer to learn a linear layer with a learning rate of 0.001 and epoch of 20.

**ACIL** (Zhuang et al., 2022b). For ACIL, we use the buffer size of 10000 for all the benchmark datasets.

**FeCAM** (Goswami et al., 2023). For backbone training of FeCAM, we use the official implementation of 200 training epochs with initial learning rate of 0.1 and batch size of 128 in the first task. Then the backbone is frozen. For the hyperparameters in FeCAM, we set $\beta = 0.1, \alpha_1 = 1, \alpha_2 = 2$.

**DS-AL** (Zhuang et al., 2024b). For DS-AL, we use the same setting of buffer size of 10000 as ACIL and tune the compensation ratio $C$ from $\{0.5, 1.0, 1.5, 2.0\}$. We use $C = 1.0, 1.5, 1.0$ for CIFAR-100, Tiny-ImageNet, and ImageNet-100 respectively.

**DPCR (ours)**. For DPCR, we use the $\gamma = 200, 2000$ and $2000$ for CIFAR-100, Tiny-ImageNet, and ImageNet-100 respectively and the $\epsilon$ is set to be 1e-9 for all experiments.

## C. Comparison with EBCIL Methods

In this section, we provide the results for comparing with EBCIL methods, including iCaRL (Rebuffi et al., 2017), PODNet (Douillard et al., 2020) and FOSTER (Wang et al., 2022). The experiments are conducted on CIFAR-100 and ImageNet-100 with the memory size $\mathcal{M} = 500, 1000$ for storing exemplars. Since these methods do not offer official implementation on Tiny-ImageNet, we skip this dataset for correctness. The experiments are conducted with PyCIL repository (Zhou et al., 2023) and we follow the official implementation of hyperparameters with the same seed of 1993.

As shown in Table. 6, even comparing with the EBCIL methods, our DPCR can have competitive performance. When the memory size is 500, all the EBCIL methods perform poorly and our DPCR outperforms them with considerable gap. With the increased memory size of 1000, the advantage of replaying exemplars begins to emerge. However, our DPCR can still achieve the second best or even best results with T=10. Since EFCIL could experience more intensive forgetting without the aid of exemplars (He et al., 2024a), these results showcases our DPCR's capability of resisting forgetting.

*Table 6.* Comparison with EBCIL methods. Results in **bold** are the best within the compared methods and data underlined are the second best under the same setting of $T$.

| Methods | Memory Size | CIFAR-100 | | | | ImageNet-100 | | | |
|---|---|---|---|---|---|---|---|---|---|
| | | T=10 | | T=20 | | T=10 | | T=20 | |
| | | $\mathcal{A}_f$ | $\mathcal{A}_{\text{avg}}$ | $\mathcal{A}_f$ | $\mathcal{A}_{\text{avg}}$ | $\mathcal{A}_f$ | $\mathcal{A}_{\text{avg}}$ | $\mathcal{A}_f$ | $\mathcal{A}_{\text{avg}}$ |
| iCaRL-CNN (Rebuffi et al., 2017) | 500 | 22.28 | 45.91 | 20.12 | 41.59 | 22.08 | 44.86 | 14.64 | 35.40 |
| iCaRL-NCM (Rebuffi et al., 2017) | 500 | 39.56 | 58.16 | 33.67 | 52.39 | 36.14 | 57.47 | 28.44 | 48.71 |
| PODNet (Douillard et al., 2020) | 500 | 31.90 | 51.17 | 23.44 | 42.42 | 38.10 | 57.10 | 25.70 | 43.56 |
| FOSTER (Wang et al., 2022) | 500 | 37.15 | 49.18 | 30.85 | 40.06 | 41.52 | 60.35 | 36.40 | 38.86 |
| iCaRL-CNN (Rebuffi et al., 2017) | 1000 | 32.33 | 53.15 | 28.45 | 49.92 | 31.78 | 52.35 | 22.56 | 43.47 |
| iCaRL-NCM (Rebuffi et al., 2017) | 1000 | 46.78 | **62.62** | 40.71 | **57.47** | 44.46 | 62.60 | 35.68 | 54.24 |
| PODNet (Douillard et al., 2020) | 1000 | 35.57 | 55.00 | 28.71 | 49.09 | 42.26 | 60.72 | 31.20 | 49.31 |
| FOSTER (Wang et al., 2022) | 1000 | 40.42 | 53.15 | **41.79** | 55.41 | **54.52** | 66.51 | **47.86** | **61.93** |
| **DPCR (ours)** | - | **49.59** | 62.13 | 40.72 | 54.91 | 53.24 | **68.18** | 40.82 | 57.94 |

# D. Extended Experiments of CN

**CN Is Beyond the Regularization.** In this section, we provide the study on CN and regularization factor $\gamma$. The regularization of ridge regression is important and affects the classification performance (Zhuang et al., 2024a). Also, since the normalization can be used to restraint the value of weights then regarded as regularization, here we conduct a joint study of $\gamma$ by and the CN module. As shown in Figure 7, without CN, the performance is benefited by adjusting the $\gamma$. The performance first increase then decrease with increasing $\gamma$. The results are consistent with those in (Zhuang et al., 2024a), where appropriate regularization addresses over-fitting then enhances performance while large $\gamma$ could lead to over regularized classifier. However, we can find that, the effect of introducing CN is beyond the regularization. The results with CN are overall good regardless of the degree of regularization (i.e., value of $\gamma$). This demonstrates that the CN is not just the regularization, it brings numerical stability for DPCR and could enhance the performance. With CN, from the Figure 7, $\gamma = 200, 2000$ and $2000$ can achieve overall good results.

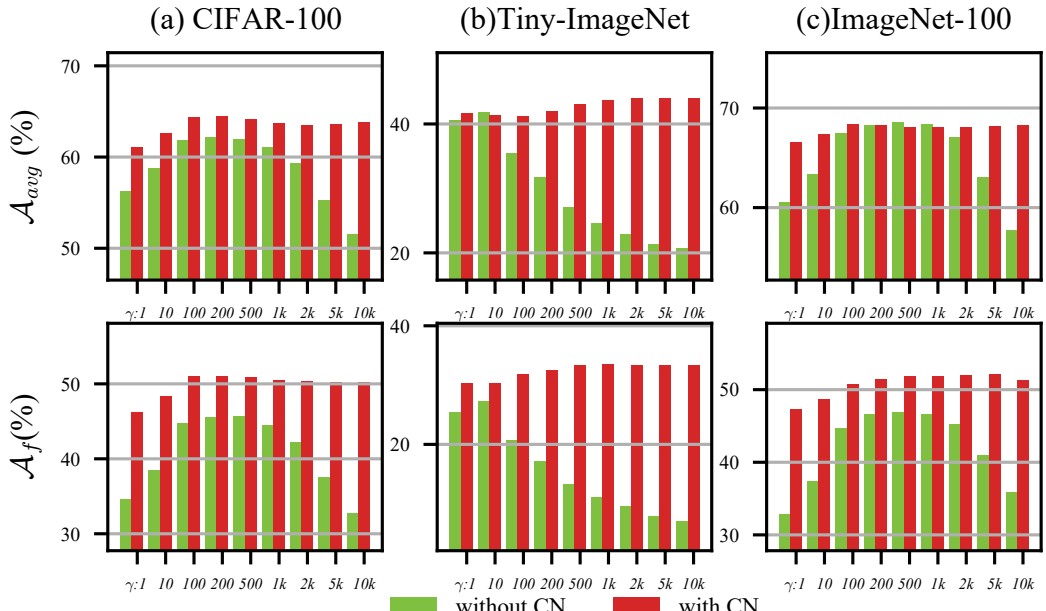

Figure 7. The effect of CN with different regularization factor $\gamma$ on three datasets.

