# OpenReview forum: "Semantic Shift Estimation via Dual-Projection and Classifier Reconstruction for Exemplar-Free Class-Incremental Learning"
_ICML.cc/2025/Conference — ICML 2025 poster_

### Official Review · Reviewer_V6Ai · 2025-02-18

**Overall Recommendation:** 3

**Summary:**

This paper focuses on Exemplar-Free Class-Incremental Learning (EFCIL), which aims to solve the problem of catastrophic forgetting without retaining any exemplars. Existing methods alleviate forgetting by storing distributional information about past tasks. However, as models are updated, past distribution information becomes outdated due to semantic shifts, and thus cannot properly represent past classes. This paper proposes a Dual-Projection Shift Estimation method to estimate and calibrate this shift. In addition, this paper uses ridge regression for offset estimation to reformulate classifier training as a reconstruction process, thereby alleviating the classifier's bias towards new categories.The proposed method achieved outstanding performance on multiple benchmark datasets.

**Claims And Evidence:**

The paper illustrates the claims made with figures and supports them by citing previous work. The paper provides experiments that demonstrate that the proposed classifier reconstruction can strike a better balance between plasticity and stability.

**Essential References Not Discussed:**

None.

**Experimental Designs Or Analyses:**

Yes. This paper follows the previous experimental design on three benchmark datasets. Experiments on larger datasets are beneficial to further demonstrate the effectiveness and generalizability of the proposed method.

**Methods And Evaluation Criteria:**

The proposed method is meaningful for the domain of Exemplar-Free Class-Incremental Learning it focuses on. Semantic shift estimation has been extensively studied in EFCIL. The method proposed in this paper further considers the semantic shift differences between different classes compared to previous work, although this entails a more complex computation.

**Other Comments Or Suggestions:**

None.

**Other Strengths And Weaknesses:**

Strengths:

1. This paper considers both semantic shifts and decision biases.
2. The proposed method achieves state-of-the-art performance on multiple datasets.

Weaknesses:

1. There have been some previous works that estimate semantic drift and feature projection [1,2,3]. The advantages of the proposed method over previous works and how it solves the limitations of previous works should be elaborated to help readers understand the novelty of the method.
2. Referring to Eq.8 and Eq.9, does the CIP need to learn and store a linear projector for each class at each subsequent incremental phases? If so, then the additional computational and storage costs incurred by these projectors should be discussed, compared to the case where they are not used. Is this going to be a significant burden on large-scale datasets?
3. The steps of using uncentered covariance and singular vector decomposition seem to result in information loss. The error caused by this information loss gradually accumulates as the number of stages experienced by the class learned earlier increases.
4. Lack of details on training, such as learning rate and weight of distillation loss, etc.

Reference:
[1] Bring evanescent representations to life in lifelong class incremental learning. CVPR 2022.
[2] Elastic feature consolidation for cold start exemplar-free incremental learning. ICLR 2024
[3] Prospective Representation Learning for Non-Exemplar Class-Incremental Learning. NeurIPS 2024.

**Questions For Authors:**

Please see weaknesses.

**Relation To Broader Scientific Literature:**

1. Compared to previous work that only focused on task-level shifts, this paper further considers the differences in semantic shifts between different categories.
2. This paper uses ridge regression to reconstruct the classifier instead of retraining the general classifier to solve the decision bias of the classifier.

**Theoretical Claims:**

I have specifically checked the correctness of the proofs for the theoretical claims in this paper.

---

> ### Author Rebuttal · Authors · 2025-03-31
>
> ## Response to Reviewer V6Ai
>
> Thank you for your constructive and detailed feedback. We provide detailed responses to your concerns below. The discussion and experiments below will be properly included in the manuscript.
>
> ### Q1: Elaborate the advantages of proposed methods over existing works [R1,R2,R3].
>
> Response: The limitations of existing works [R1,R2,R3] can be summarized as follow respectively. 1. Toldo et al [R1] propose to train multi-layer perceptrons or variational autoencoders to estimate the semantic shift, intensifying the training cost.  EFC [R2] utilizes the Gaussian kernels to capture the shift by estimating the translation of feature means, potentially neglecting other types of shift transformations including scaling. PRL [R3] proposes a feature projection to align the current embedding space to a pre-defined latent space, rather than estimating the semantic shift. Also, PRL does not consider different classes within a training task. Our DPCR provides a DP to estimate the semantic shift in both efficient and effective way via a linear projection can be calculated directly without iterative BP-training. Also, the CIP in DP includes class-specific knowledge. To demonstrate the effectiveness of DPCR, we further include the experiments compare with EFC and PRL (we did not find the released code of [R1]). The results show the high performance of our DPCR.
>
> |CIFAR-100|T=10|| T=20||
> |-|-|-|-|-|
> |(%)|A_last|A_avg|A_last|A_avg|
> |EFC|45.01|59.44|34.12|48.63|
> |PRL|45.32|57.08|28.43|41.98|
> |**DPCR**|**49.59**|**62.13**|**37.79**|**54.48**|
>
> ### Q2: Does CIP need to learn and store a linear projector for each class at each subsequent incremental phase?
> Response: CIP is a training-free projection to include the class-specific knowledge and it does not need to be stored. Actually, during the training of DPCR, CIP is calculated from the uncentered covariance via SVD (see Eq. (6) and (7)) and it will be used to calibrate the old information set with TSSP and then discarded. Only the covariance and prototype  ${\hat{{\Phi}}{c}^{\theta{t}}, \hat{{\mu}}{c}^{\theta{t}}}$ need to be stored in DPCR. For the operations in Eq. (8) and Eq (9), they are used to demonstrate how the old features of far previous task are calibrated of without loss of generality and this process does not be necessarily performed in the implementation of DPCR. In our implementation, the calibration is conducted across two adjacent tasks (see Algorithm 1). We will further clarify this to avoid potential misunderstanding.
>
> ### Q3: Using SVD seems to result in information loss.
>
> Response: CIP is used to inject class-specific information via projecting the projector in TSSP to significant directions. The construction of projector for CIP maintains all the singular vectors corresponding to non-zero singular values and only discards the information of null space, which we believe not necessarily needed. To validate the effect of information of null space, we perform an experiment on CIFAR-100 with all the singular vectors to construct the projector of CIP (denoted as DPCR-wn). The results are shown below and we find that including null space information does not affect the performance and thus the information loss may not have large impact on the performance.
>
> |CIFAR-100|T=10|| T=20||
> |-|-|-|-|-|
> |(%)|A_last|A_avg|A_last|A_avg|
> |DPCR-wn|49.59|62.14|38.04|54.25|
> |DPCR|49.59|62.14|38.04|54.25|
>
> ### Q4: Lack of details of training.
>
> Response: As indicated in line 310 (left), the implementation details are provided in Appendix B. We will move them to the manuscript to avoid possible omission.
>
> ### Q5: Experiments on larger datasets are beneficial.
>
> Response: As suggested, we conduct the validation on ImageNet-1000 with T=10 using ResNet-18 under the same seed of 1993. The hypeparameters are the same as those in ImageNet-100 experiments. The results shows our DPCR is still competitive on ImageNet1000.
>
> |ImageNet-1k(T=10)|A_last|A_avg|
> |-|-|-|
> |LwF|22.01|42.40|
> |ACIL|32.28|46.61|
> |DSAL|33.67|48.84|
> |LDC|35.15|53.88|
> |ADC|31.34|50.95|
> |**DPCR**|**35.49**|**54.22**|
>
> [R1] Bring evanescent representations to life in lifelong class incremental learning. CVPR 2022.
>
> [R2] Elastic feature consolidation for cold start exemplar-free incremental learning. ICLR 2024
>
> [R3] Prospective Representation Learning for Non-Exemplar Class-Incremental Learning. NeurIPS 2024.

---

### Official Review · Reviewer_YwAB · 2025-02-22

**Overall Recommendation:** 3

**Summary:**

This paper proposes a method called Dual-Projection Shift Estimation and Classifier Reconstruction (DPCR) to solve two key challenges in Exemplar-Free Class-Incremental Learning (EFCIL): semantic shift and decision bias. It reconstructs the classifier through the dual projection mechanism and ridge regression, effectively balancing the learning of new and old knowledge. Experimental results show that DPCR outperforms the existing EFCIL method on multiple benchmark datasets.

**Claims And Evidence:**

This paper claims to address semantic shift via proposed DP, and decision bias via RRCR. The claims are supported by the experiments.

**Essential References Not Discussed:**

No

**Experimental Designs Or Analyses:**

The experiment designs and analyses are adequate. But there are figures lacking of horizontal labels (Figure 4).

**Methods And Evaluation Criteria:**

The methods are illustrated clearly. The evaluation criteria is common. A slight drawback is the lack of the evaluation on large datasets like ImageNet1000. Besides, this paper lacks evaluation on the complexity of the method.

**Other Comments Or Suggestions:**

Overall, this paper proposes a novel method in NECIL, and seems achieving promising results with reasonable approaches. Although with some drawbacks, the paper seems good.

**Other Strengths And Weaknesses:**

No

**Questions For Authors:**

No

**Relation To Broader Scientific Literature:**

No

**Theoretical Claims:**

There are no new theoretical results.

---

> ### Author Rebuttal · Authors · 2025-03-31
>
> ## Response to Reviewer YwAB
>
> Thank you for your constructive and detailed feedback. We provide detailed responses to your concerns below. The discussion and experiments below will be properly included in the manuscript.
>
>
> ### Q1: Lack of evaluation on lager datasets like ImgaNet1000.
> Response: As suggested, we validate DPCR and compared methods on ImageNet1000 with T=10. We use the same backbone of ResNet-18 and same under the same seed of 1993 to conduct the experiments. The hypeparameters are adopted the same as those in ImageNet-100 experiments. The results are reported below and our DPCR is still competitive on ImageNet1000. These results will be included in the manuscript.
>
> |ImageNet-1k(T=10)|A_last|A_avg|
> |-|-|-|
> |LwF|22.01|42.40|
> |ACIL|32.28|46.61|
> |DSAL|33.67|48.84|
> |LDC|35.15|53.88|
> |ADC|31.34|50.95|
> |**DPCR**|**35.49**|**54.22**|
>
> ### Q2: Lack of complexity analysis
> Response: As suggested, we will include the complexity analysis. Since the incremental representation learning is common in the CIL realm, here we analyze the complexity of DP and RRCR. The complexity analysis will be included as follow.
>
> Suppose $F$ is the FLOPs of the model ($F \approx 1.8\times10^{9}$ for ResNet-18),  the time complexity of DP $\mathcal{O}(FN_t + N_td^2 + tC(d^3 + C^2d))$ is the sum of those of the feature extraction $\mathcal{O}(FN_t)$, calculation of the task-wise projection matrix $\mathcal{O}(N_td^2+d^3)$, and the rectification of each class $\mathcal{O}(d^3+C^2d)$. Similarly, the time complexity of RRCR is $\mathcal{O}(d^2N_c + dtC + td^2+d^3)$.
>
> ### Q3: Some figures lack of horizontal labels.
> Response: Thank you for pointing out this. We will check and include the horizontal labels for all the figures.

---

### Official Review · Reviewer_spyP · 2025-03-12

**Overall Recommendation:** 2

**Summary:**

The paper studies exemplar-free class-incremental learning and focus on addressing two major problems of semantic shift and decision bias. The authors propose to use dual-projection to estimate semantic shift which includes both class-wise and task-wise shifts, and a ridge regression based classifier instead of a learnable classifier to address the problem of decision bias. The proposed method achieves better stability-plasticity trade-off among existing methods in EFCIL benchmarks.

**Claims And Evidence:**

Most claims are validated.
Some claims need better explanation and validation:
1. Why using Ridge Regression-based Classifier Reconstruction is better than other methods like NCM or FeCAM [X] which also employs covariance based classification? What is the motivation for introducing this RR based classifier? Does RRCR improve over Mahalanobis-distance based classification as done in [X] for EFCIL?
2. Line 066-069: “However, its effectiveness heavily depends on the quality of the learned representations, which are susceptible to degradation caused by semantic shift.” This motivation is neither clear nor justifies introducing a new type of classifier. Existing methods like LDC, ADC exactly solve the semantic shift problem. I would ask the authors to justify why it is required to introduce a Ridge Regression-based Classifier in the context of EFCIL and what are the benefits of using this.

[X] FeCAM: Exploiting the heterogeneity of class distributions in exemplar-free continual learning. In Advances in Neural Information Processing Systems, 2023.

**Essential References Not Discussed:**

1. Most essential references are discussed.
2. Some discussion on FeTrIL [WACV 23] would be useful in the context of the paper since they also propose an alternative SVM-based classifier instead of training linear classifier. Comparison with a recent work FeCAM [NeurIPS 23] could be added which also uses covariance of features for classification.
3. It is important to acknowledge that learning a projector for estimating semantic shift in prototype means was also proposed by LDC (ECCV 24). While the proposed method builds on top of this concept, the authors do not acknowledge this in the paper.
4. While the paper focus largely on semantic shift, it does not have much discussion on existing drift compensation or resistant methods like LDC, ADC. Adding some discussion on this to better understand existing methods solving the same problem would add more clarity. Why these methods are not good enough or how is the proposed method addressing their limitations? This would improve the motivation of the proposed method.
5. References to ridge regression theory is missing.

**Experimental Designs Or Analyses:**

The experimental setup is sound and valid, and following the standard practice in EFCIL. The analysis and ablation studies are extensive.

**Methods And Evaluation Criteria:**

The evaluation criteria is appropriate for the EFCIL setting. The compared methods are very relevant and recent.
Additionally, I would propose to include evaluation on some fine-grained classification datasets similar to what is done in ADC [CVPR 24] for a more rigorous experimentation.

**Other Comments Or Suggestions:**

Writing mistakes:
1. Line 189 (right side) - “Then we can constructed the projector of CIP as”.

**Other Strengths And Weaknesses:**

Strengths
The paper is well written and organized, with extensive experiments and ablations, good illustrations.

Weaknesses:
Discussed above. My main concern is that the motivation for using Ridge Regression-based Classifier Reconstruction is not convincing. More discussions on "why use RRCR" is important.
Some more comparisons, discussion and validation as mentioned above will improve the paper.

I am willing to increase my rating if the authors address my major concerns.

**Questions For Authors:**

1. How is decision bias different from task-recency bias which is discussed in many CL works? It is better to stick to existing terms instead of defining the same thing with a different term.

**Relation To Broader Scientific Literature:**

The contributions of this paper aims to address problems of semantic shift and task-recency bias discussed in several existing CL works.

**Theoretical Claims:**

I do not find any errors in the theoretical formulation.
Minor suggestion: It is important to mention that the covariance matrix referred to in equation 15 is not centered on the mean and also not normalized. This can be referred to as gram matrix as done in [Y].

[Y] Ranpac: Random projections and pre-trained models for continual learning. Advances in Neural Information Processing Systems. 2023

---

> ### Author Rebuttal · Authors · 2025-03-31
>
> ## Response to Reviewer spyP
> Thank you for your constructive feedback. We provide detailed responses to your concerns below. The discussion and experiments will be properly included. Also, we will check and revise all the writing mistakes, and include references of ridge regression.
> ### Q1: Comparison between RRCR and NCM/Mahalanobis-distance classifier in FeCAM.
> Response:  RRCR is a parametrized classifier where the decision boundaies can be directly modified by training the parameters. NCM and Mahalanobis-distance classifiers are non-parametric classifiers that utilizes feature distribution and absence of trainable parameters in limits their adaptability across task. Although they can avoid decision bias and achieve better performance over existing parametrized classifiers, RRCR mitigates this via reconstruction then keep the benefit as a parametrized approach. To validate this, we compare RRCR with NCM and Mahalanobis on CIFAR-100 (T=10) using the same frozen backbone after task 1 (following FeCAM). RRCR surpasses both.
>
> |CIFAR-100(T=10)|A_last(%)|A_avg(%)|
> |-|-|-|
> |FeCAM|30.41|45.59|
> |NCM|18.77|36.16|
> |**RRCR**|**33.28**|**49.16**|
>
> RRCR also better adapts to semantic shift during CIL via reconstruction. FeCAM freezes the backbone to maintain consistent covariances, but this impedes plasticity. Once the backbone evolves, prior class covariances become outdated and hard to update without storing previous data due to centralization to obtain covariance. RRCR, by contrast, recalibrates using ${\hat{{\Phi}}{c}^{\theta{t}}, \hat{{\mu}}{c}^{\theta{t}}}$ via matrix operations, avoiding the need to retain prior data or embeddings.
> ### Q2: Motivation of RRCR.
> Response: In this paper, we propose RRCR to address decision bias in parametrized classifier during CIL. Non-parametric classifiers like NCM lack lack adaptability due to their static structure and inability to directly modify decision boudaries. While techniques like ADC/LDC try to offset semantic shift, they remain limited by non-learnable classifiers. Existing parametrized classifiers can directly learn from labels but tend to become biased without old data. RRCR fills this gap, offering the benefits of a learnable model while correcting bias via reconstruction. We will clarify this motivation in lines 066–069.
> ### Q3: Evaluation on fine-grained datasets.
> Response: As suggested, we include the results on fine-grained dataset CUB-200 with T=5 in cold-start setting with the same seed. Results shows that DPCR also leads the compared methods.
> |CUB200(T=5)|A_last(%)|A_avg(%)|
> |-|-|-|
> |LwF|25.40|36.38|
> |ACIL|21.14|33.14|
> |DSAL|21.28|32.36|
> |SDC|24.21|36.00|
> |LDC|28.70|39.09|
> |ADC|28.84|39.44|
> |**DPCR**|**29.51**|**40.62**|
> ### Q4: Covariance vs. Gram Matrix.
> Response: We will replace "uncentered covariance" to "gram matrix".
> ### Q5: Compare with FeCAM.
> Response: We compare DPCR with FeCAM and obtain the results of FeCAM via the official implementation under the same seed of 1993. Our DPCR outperforms FeCAM and this can be attributed to that FeCAM needs to freeze backbone thus limits the plasticity. Our DPCR can take advantage of incremental representation learning with semantic shift estimation then a good stability-plasticity balance.
> |CIFAR-100|T=10||T=20||
> |-|-|-|-|-|
> |(%)|A_last|A_avg|A_last|A_avg|
> |FeCAM|34.82|49.14|25.77|41.21|
> |**DPCR**|**50.08**|**63.43**|**41.62**|**56.01**|
> |Tiny-ImageNet|T=10||T=20||
> |(%)|A_last|A_avg|A_last|A_avg|
> |FeCAM|29.83|42.19|22.69|34.48|
> |**DPCR**|**35.50**|**47.73**|**26.67**|**38.80**|
> |ImageNet-100|T=10||T=20||
> |(%)|A_last|A_avg|A_last|A_avg|
> |FeCAM|41.92|58.21|28.64|43.04|
> |**DPCR**|**53.46**|**68.20**|**40.76**|**57.81**|
> ### Q6: Acknowledge LDC and add discussion of LDC, ADC, and FeTrIL.
> Response: We will include the discussion as follow. To estimate the semantic shift, ADC uses adversarial samples to estimate the translation of prototypes, and LDC introduces a learnale projection. However, ADC only considers the shift of translation, neglecting other transformation including scaling, and LDC only estimates the shift across tasks without class-specific information. Moreover, LDC needs iterative BP-training to get the projector, inducing more computation cost. Inspired by the learnable projector in LDC, DPCR achieves shift estimation via DP with low-cost closed-form solution and class-specific information. FeTrIL uses prototype-based pseudo-features to rebalance classifier training and employs a LinearSVC. However, it freezes the backbone, reducing plasticity. Also, LinearSVC's effect in CIL is not thoroughly studied.
> ### Q7: Decision bias vs. Task-recency bias?
> Response: Task-recency bias refers to the model's tendency to favor new tasks. Several works attribute it to the bias in classifier. We use “decision bias” to clarify that the bias originates in the classifier. While similar in outcome, we prefer the more specific term **decision bias** for better understanding and will clarify this in the text.

---

> > ### Comment · Reviewer_spyP · 2025-04-06
> >
> > I thank the authors for their detailed response to the questions. Most of my concerns are addressed.
> >
> > However, the motivation of using RRCR is still not well-formed. I do not agree with the authors rebuttal statement "RRCR is a parametrized classifier where the decision boundaies can be directly modified by training the parameters". Can the authors specify what exactly are the trainable parameters in the RRCR classifier?
> >
> > As far as I understand, the model feature extractor is trainable in new tasks unlike in NCM or Fecam, but the RRCR classifier is not trainable and also a non-parametric classifier which simply uses the feature distributions. Why is RRCR referred to as a parametrized classifier? This is still not clear to me and more clarifications are required.

---

> > > ### Author Response · Authors · 2025-04-06
> > >
> > > Thank you for taking time to read our response! We apologize that our delivery might lead to misunderstanding that RRCR is non-parametric since the formulation of RRCR is **very different** from the training of existing parametrized classifiers. We provide the clarification as follows.
> > >
> > > As indicated in line 223, RRCR constructs a **trainable classifier parametrized by $W_t$** and the parameters can be **learned via the least-square solution in Eq. 12**. The classifier in RRCR is basically a **fully connecting network without bias** and the forward process can be formulated as $Y=W_{t}X$. The objective function of training classifier in RRCR is the ridge regression in Eq. 11, and the optimal solution can be obtained analytically without gradient-based optimization on the classifier. To modify the training in Eq. 11 to each task in CIL, we further decompose the solution to a category-wise form (Eq. 13), and this may lead to misunderstanding that RRCR is simply based on the feature distributions. However, the solution is originated in the training result of the objection function Eq. 11.
> > >
> > > Eq. 11: $\underset{W_{t}}{\operatorname{argmin}}~\lVert Y_{1:t} - X_{1:t}^{\theta_1} W_{t}\rVert_{\text{F}}^{2} + {\gamma}\lVert W_{t}\rVert_{\text{F}}^{2}$
> > >
> > > Eq. 12: $\hat{W_t} = (\sum_{i=1}^{t} X_{i}^{\theta_{t}\text{T}} X_{i}^{\theta_{t}}+ \gamma {I})^{-1}\sum_{i=1}^{t} X_{i}^{\theta_{t}\text{T}} Y_i$
> > >
> > > Eq. 13: $\hat{W_t} = (\sum_{i=1}^t \sum_{c \in \mathcal{C}_i}^{|\mathcal{C}_i|} \Phi^{\theta_t} _ {i,c} + \gamma {I})^{-1} \sum^t _ {i=1} \sum _ {c \in \mathcal{C}_i}^{|\mathcal{C}_i|} H^{\theta_i} _ {i,c}$
> > >
> > > The training of classifier in RRCR could differ from backpropagation in existing classifier training significantly and potentially lead to misunderstanding. We sincerely apologize for any lack of clarity in our previous presentation of RRCR and will provide further explanations to address potential confusion.

---

### Decision · Program_Chairs · 2025-05-01

**Decision:**

Accept (poster)

**Comment:**

The paper presents an approach for exemplar-free class-incremental learning that simultaneously addresses the challenges of semantic shift and decision bias. The paper received mixed reviews, with primary concerns focusing on missing results for fine-grained datasets, lack of comparison with existing methods, and an unclear motivation for the Ridge Regression-based Classifier Reconstruction (RRCR) component. Several of these issues were clarified during the rebuttal phase; however, one reviewer further requested other clarification regarding RRCR, which was subsequently provided by the authors. The AC has reviewed both the paper and the rebuttal and agrees that all major concerns have been satisfactorily addressed. As such, the paper can be accepted.